# Impact of Storm Surge on the Yellow River Delta: Simulation and Analysis

**Liang Huang** [1], **Shenliang Chen** [1,*], **Shunqi Pan** [1,2], **Peng Li** [1] and **Hongyu Ji** [1]

1   State Key Laboratory of Estuarine and Coastal Research, East China Normal University, Shanghai 200241, China

2   Hydro-Environmental Research Centre, School of Engineering, Cardiff University, Cardiff CF24 3AA, UK

*   Correspondence: slchen@sklec.ecnu.edu.cn; Tel.: +86-021-54836498

**Abstract:** Storm surges can lead to serious natural hazards and pose great threats to coastal areas, especially developed deltas. Assessing the risk of storm surges on coastal infrastructures is crucial for regional economic development and disaster mitigation. Combining in situ observations, remote sensing retrievals, and numerical simulation, storm surge floods in the Yellow River Delta (YRD) were calculated in different scenarios. The results showed that NE wind can cause the largest flooding area of 630 km$^2$, although the overall storm surge risk in the delta is at lower levels under various conditions. The coastal oilfields are principally at an increasing storm surge risk level. E and NE winds would result in storm surges of 0.9–1.4 m, increasing the risk of flooding in the coastal oilfields. Nearshore seabed erosion in storm events resulted in a decrease in inundation depths and inundation areas. To prevent and control storm surge disasters, we should adapt to local conditions. Different measures should be taken to prevent the disaster of storm surges on different seashores, such as planting saltmarsh vegetation to protect seawalls, while the key point is to construct and maintain seawalls on high-risk shorelines.

**Keywords:** storm surge; Yellow River Delta; risk assessment; coastal oilfield

## 1. Introduction

Storm surge is a globally common marine natural disaster that strikes coastal areas, especially estuaries, bays, and deltas. It not only destroys ports, docks, and embankments but also inundates houses, farms, and aquaculture areas after the breach of coastal defense has been damaged, causing fatalities and serious economic losses [1,2]. Between 1949 and 2016, the global direct economic losses caused by storm surges reached 495.2 billion USD [3]. In China alone, there have been 397 storm surges since 2000, resulting in 888 missing or dead people and CNY 220.492 billion in direct economic losses (http://www.mnr.gov.cn, accessed on 20 April 2022). In the future, the frequencies and destructiveness of storm surge disasters will increase significantly as global warming intensifies relatively to sea level rise [4].

The modern Yellow River Delta (YRD) is located in the storm surge hazard area around the Western Pacific [5]. Owing to meteorological, topographical, and hydrological factors, the YRD was frequently threatened by severe storm surges [6]. From 2014 to 2021, a total of 42 storm surge events were reported in the YRD. Especially in August 2019, super typhoon 'Likema' caused serious damage to local aquaculture, infrastructure, and embankments, with direct economic losses of CNY 2.163 billion (https://www.mnr.gov.cn, accessed on 20 April 2022). Therefore, it is of great significance to improve the comprehensive risk assessment system for storm surges and enhance the ability of storm surge prediction and warning capability. Previous research has mainly focused on the aspects of storm surge simulations [7], geomorphologic evolution [8,9], and disaster mechanisms [6], while relatively little research has been conducted on storm surge risk assessment.

Based on regional disaster system theory, the risk assessment of storm surge adopted storm surge hazard and vulnerability to assess disaster risks [10,11]. Risk assessment mainly refers to the assessment of the natural elements of storm surges, usually through numerical simulations and recurrence period calculations. Vulnerability refers to the vulnerability of hazard-affected bodies to certain disasters, reflecting their ability to withstand storm surges [12]. Therefore, numerical simulation methods such as flood routing, study area inundation analysis, and comprehensive vulnerability analysis of the hazard-bearing body can effectively evaluate storm surges. The storm surge risk map drawn on this basis can effectively indicate the susceptible areas, enhance disaster preparedness, and greatly mitigate the impact of storm surges [13]. This storm surge risk assessment method has been widely used [12–14], and its effectiveness has been verified in local disaster prevention and mitigation practices.

In this study, we simulated storm surge flooding in the YRD in different scenarios with numerical calculations, gave the flood area and risk classification with high spatial resolution, explained the mechanism of storm surge flooding in the YRD, discussed the influence regions of storm surges and mapped the risk level of storm surge under different weather conditions. The research findings would provide a scientific reference for storm surge prevention and mitigation in delta areas.

## 2. Material and Methods

### 2.1. Study Area

The modern Yellow River Delta (YRD) has been developed since 1855 when the Yellow River emptied into the Bohai Sea (Figure 1a). It is located between Bohai Bay and Laizhou Bay, with an area of 5450 km$^2$ and a population of about 2 million. The YRD is mainly composed of four towns, namely Diaokou, Xianhe, Gudao, and Huanghekou. As a key industrial zone in China, the YRD is a site of oil and gas exploration, salt production and aquaculture. Since 1985, a series of coastal defenses have been built from the east to the northeast of the YRD (Figure 1c), including the Gudong Seawall, Dongying Port, and Groyne, most of which are located above 3.5 m elevation [15].

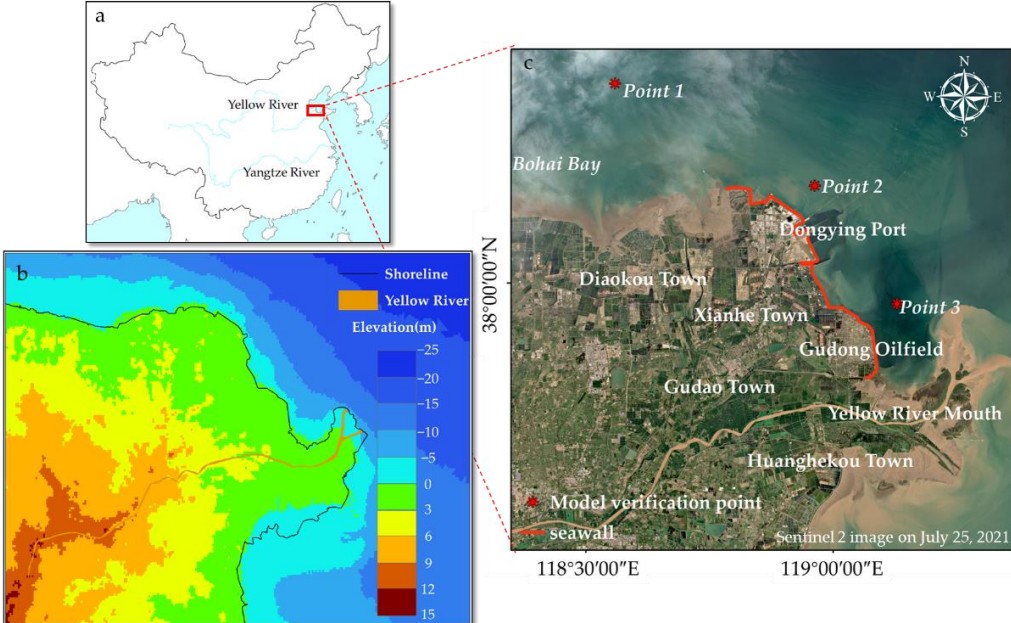

**Figure 1.** Map of the study area: (**a**) location of the Yellow River Delta (YRD); (**b**) elevation of the delta; and (**c**) satellite images of the YRD region.

The YRD, adjacent to the Bohai Sea, has a warm temperate semi-humid continental monsoon climate with a mean annual temperature of 13.2 °C, precipitation of 540.3 mm,

and wind speed of 2.8 m/s. The nearshore tides are dominated by irregular semidiurnal tides, with an average tidal range of 0.6 to 0.8 m [16]. Waves in the study area are generally generated by the southerly winds in summer and the NE winds in winter [17].

The YRD has an open topography with a gentle slope and high west and low east. The average elevation in the southwest is 11 m, and the peak is located at 13.3 m south of Lijin. The regional natural gradient is 1/8000~1/12,000 and is extremely vulnerable to storm surges (Figure 1b).

### 2.2. Data Collection and Pre-Processing

The dataset used in this study mainly includes topo-bathymetric, wind field, and hydrological data (Table 1). The distance between bathymetric cross-sections is approximately 1000 m, and the distance between adjacent measuring points in a cross-section is about 250 m. The data plane coordinate is Beijing 54 and the elevation system is 1956 Yellow Sea Elevation. The Bohai bathymetric plane is World Geodetic System (WGS) 84, and the elevation employs the 1985 National Height Datum. The plane coordinate system of General Bathymetric Chart of the Oceans (GEBCO) global Digital Elevation Model (DEM) data and Shuttle Radar Topography Mission Digital Elevation Model (SRTMDEM) 90 m topographic data is WGS 84, and the elevation is derived from the geoid-based elevation data. To reduce measurement error, the average value of the DEM data was aggregated. The coordinate system of topo-bathymetric data was unified into WGS 84 as the ArcGIS platform, and the elevation was unified as the 1985 National Height Datum. The conversion method between each elevation is as follows:

$$1985 \text{ National Elevation} = 1956 \text{ Yellow Sea Elevation} - 0.029 \text{ m} \tag{1}$$

$$1985 \text{ National Elevation} = \text{Geoid-Based Elevation Datum} + 0.21 \text{ m} \tag{2}$$

**Table 1.** Source of included data.

| Data Type | Data Items | Source |
|---|---|---|
| Topo-bathymetric data | Terrestrial topography<br>Nearshore bathymetry<br>Bohai Sea bathymetry<br>Bathymetry outside of Bohai Sea | Geospatial Data Cloud (SRTMDEM 90 m)<br>Measured sectional bathymetry (1985, 2015)<br>5 × 5 km resolution of the Bohai Sea bathymetric data (908 Project)<br>National Marine Data Center (GEBCO Global 15″ Resolution DEM Data) |
| Wind field data | Magnitude, cause, and duration of storm surges<br>Wind field data during storm surges | Dongying Municipal Bureau of Marine Development and Fisheries (http://hsdy.dongying.gov.cn/, accessed on 20 April 2022)<br>European Centre for Medium-Range Weather Forecasts (EAR5 Reanalysis Data) |
| Hydrological data | Flow velocity and direction data<br><br>Hydrology during storm surges | 27 h of hydrological observation outside Dongying Port and Gudong nearshore in September 2021 (Point 1 and Point 2 in Figure 1c)<br>Observation data from a bottom tripod were used outside the Diaokou estuary in February 2020 (Point 3 in Figure 1c) |
| Satellite image | Sentinel 2 image on 25 July 2021 | European Union Aviation Safety Agency |

### 2.3. Hydrodynamic Model Settings

From 2014 to 2021, 21 (50%) of the 42 recorded storm surge events in the YRD were caused by cold air alone. Another 11 events, 26.19%, were caused by the combined effects of cold air and low pressure, 6 by typhoons, and 4 by extratropical cyclones. Figure 2 records the time, causes, and water level of the storm surges in the YRD from 2014 to 2021, indicating the seasonal and frequent occurrence of storm surges in the area from February to April and from August to November. Cold air often leads to stronger storm surges, particularly the one caused by the 2019 typhoon 'Likema', which reached 190 cm and emerged as the largest surge since 2014.

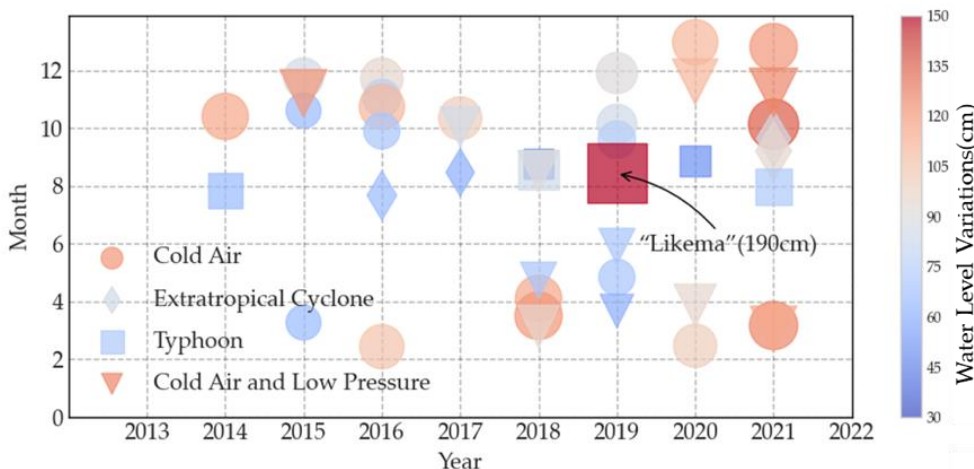

**Figure 2.** Monthly distribution of storm surges of different causes in the YRD from 2014 to 2021.

The primary driver of storm surges is wind stress. By analyzing the wind direction of 42 storm events in Dongying City, it was found that the main wind directions are NE, N, and E, accounting for 52.38%, 23.81%, and 11.90%, respectively. According to the observation at Dongying Port Marine Station from 2011 to 2015 [18], the strong wind in Dongying Port mainly occurs in the NE, NW, N, W, and E directions. In the NE direction, the average extreme wind speed is 23.02 m/s, with a peak value of 27.90 m/s. The maximum wind speed in the N direction is 28.2 m/s with an average extreme wind speed of 24.24 m/s; the maximum wind speed in the E direction is 30.6 m/s, while the average extreme wind speed is 23.44 m/s. In the work of [7], Mesoscale Model 5 (MM5) was used to simulate the hourly wind field in the YRD, and Pears-III distribution was used to calculate the wind speed return period in five directions, as shown in Figure 3.

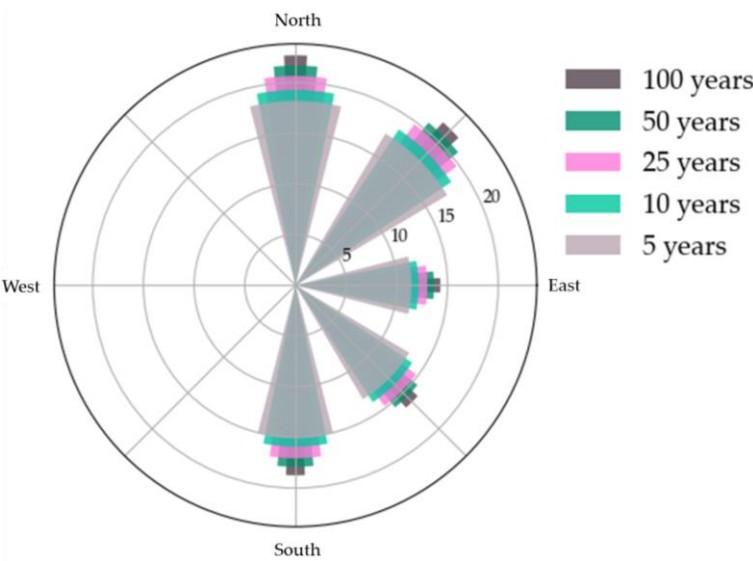

**Figure 3.** Speed of different return period winds in five directions in the YRD (unit: m/s).

In this study, MIKE 21, a two-dimensional free surface simulation system developed by Danish Hydraulic Institute (DHI), was used to construct the YRD hydrodynamic model, as it is suitable for water quality and hydraulic simulation of the estuary, bay, lake, and other areas. The computational domain is discretized by the finite volume method, and the two-dimensional unsteady flow of vertically homogeneous fluid is simulated based on the time-averaged Navier–Stokes equations [19]. The simulation model covers the entire Bohai, the Yellow Sea, and the northern part of the East China Sea (Figure 4). The model mesh contains all the land in the YRD and is refined near the coastline, with a total of

77,984 nodes and 76,795 cells. Unstructured grids are used for all cells, and the cell size is proportional to the distance from the YRD shoreline. The smallest cell is located in the Yellow Estuary with a length of 200 m, while the largest unit is located in the East China Sea, with a side length of 10,000 m. The model has only an open boundary in the south of the domain, where the tidal level boundary condition is imposed. The tide level provided by the software is corrected using Equation (2). At sea, the Manning roughness coefficient increases with the increase in water depth, ranging from 0.04 to 0.007, while on land, the Manning roughness coefficient is set to 0.06. Considering the wind field, the wind drag coefficient is set to 0.0027. The defense project of Gudong–Dongying Port is mainly built on the embankment with an elevation of 3.5 m, regardless of the breach of the sea wall.

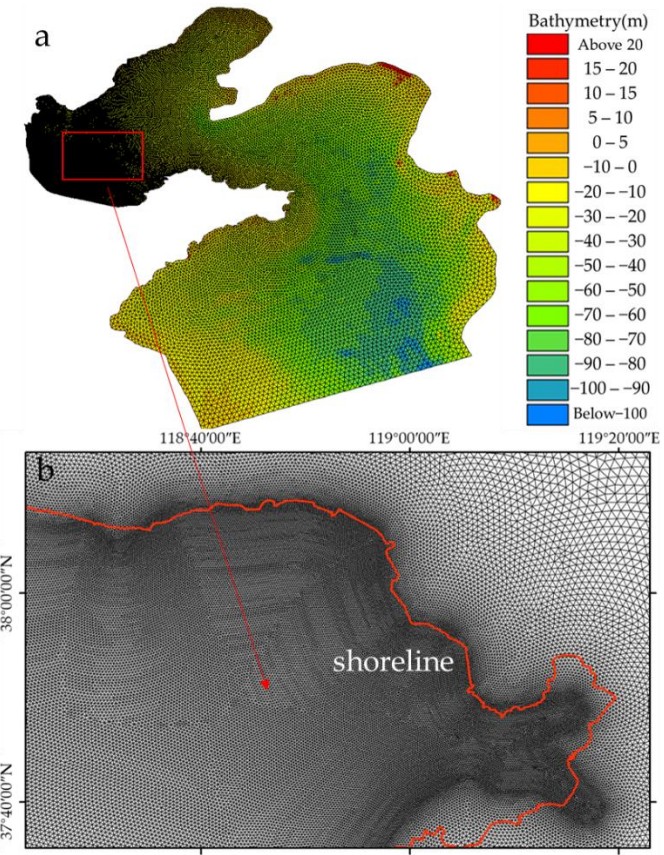

**Figure 4.** Mesh of computing domain ((**a**): model computing mesh; (**b**): delta position refinement grid).

As mentioned above, since storm surges in the YRD were largely driven by cold air fronts [7], the experimental condition was designed to be strong winds in N, NE, and E directions with 10-, 50-, and 100-year return periods. The simulation model starting at 0:00 on 25 August 2021 featured the wind field simulation at 0:00 on 1 September 2021.

### 2.4. Ground Classification and Risk Assessment System of Storm Surges

The ground objects in the YRD mainly include bare flats, roads, vegetation, water ponds, and buildings. The Sentinel Application Platform (SNAP) is used for atmospheric correction of the European Space Agency (ESA) Sentinel 2 image. Ground objects are identified using normalized difference water index (NDWI), normalized difference vegetation index (NDVI), and visual image interpretation. The specific methods are as follows.

Based on the fact that water has the strongest absorption and almost no reflection in the near- and mid-infrared wavelength range [20], the NDWI is calculated by comparing visible and near-infrared band highlights to monitor water information in the images. The expression is:

$$NDWI = (Green - NIR)/(Green + NIR) \tag{3}$$

where Green denotes the green band, and NIR represents the near-infrared band. The threshold value is set to 0, i.e., any index value greater than 0 indicates a water body.

NDVI is the ratio parameter of near-infrared (NIR) and infrared (R) reflectance in remotely sensed images. As the most widely used index to characterize vegetation cover [21], NDVI performs well in characterizing bare flats and rock. The expression is:

$$NDVI = (NIR - Red)/(NIR + Red) \tag{4}$$

where Red denotes the red band, and NIR represents the near-infrared band. By setting the index threshold to 0.2, 0.2, NDVI > 0.2 corresponds to vegetation coverage area, and $-0.2 < NDVI < 0.2$ corresponds to a bare plain.

Using the inundation depth of the flooding area as an indicator, the conditions can be classified into four risk levels (Table 2). Both qualitative and quantitative assessments are feasible when assessing the vulnerability in real situations, and the type of land use can be used as an indicator in qualitative assessment (Table 3) [22].

**Table 2.** Risk levels of inundation depth [22].

| Risk Level | Inundation Water Depth (cm) |
|------------|------------------------------|
| I          | [300, +∞)                    |
| II         | [120, 300)                   |
| III        | [50, 120)                    |
| IV         | [15, 50)                     |

**Table 3.** Vulnerability of different land use status [22].

| Current Status of Land Use | Ranges of Vulnerability | Vulnerability Level |
|----------------------------|--------------------------|----------------------|
| Cultivated land            | 0.1~0.2                  | IV                   |
| Meadow and woodland        | 0.1                      | IV                   |
| Industrial and storage land| 0.6~1                    | II~I                 |
| Residential land           | 1                        | I                    |
| Water pond                 | 0.3                      | IV                   |
| Bare flat                  | 0.1                      | IV                   |
| Facility agricultural land | 0.2~0.5                  | IV~III               |

Based on the results returned by the risk and vulnerability assessment, storm surge risk assessment can be calculated by Equation (5) [12].

$$R = H \times V \tag{5}$$

where, R, H, and V represent risk, hazard, and vulnerability, respectively. $\times$ denotes the risk level recognition matrix (Table 4).

**Table 4.** Storm surge risk levels in relation to hazard and vulnerability levels [22].

|              |              | Vulnerability Levels | | | |
|--------------|--------------|-----------------------|-----------------------|-----------------------|----------------------|
|              |              | Low (IV) [0.1,0.3]    | Lower (III) (0.3, 0.5]| Higher (II) (0.5, 0.8]| High (I) (0.8, 1.0]  |
| Hazard levels| Low (IV)     | Low risk (IV)         | Low risk (IV)         | Lower risk (III)      | Lower risk (III)     |
|              | Lower (III)  | Low risk (IV)         | Lower risk (III)      | Higher risk (II)      | Higher risk (II)     |
|              | Higher (II)  | Lower risk (III)      | Higher risk (II)      | Higher risk (II)      | High risk (I)        |
|              | High (I)     | Lower risk (III)      | Higher risk (II)      | High risk (I)         | High risk (I)        |

## 3. Results

### 3.1. Verification of Numerical Model

In February 2020, 20 km from the Diaokou estuary, the TGR-2050 tide gauge (manufactured by Richard Brancker Research Company, Ottawa in Canada) was fixed in an anti-tug net and placed on the seabed (Point 1 in Figure 1c). Subsequently, the water level was observed every 10 min and corrected by a tidal station at Dongying Port. To verify the simulations of the numerical model during the storm surge, data related to the storm surge that occurred in the YRD on 14 February 2020 were utilized, with an average storm surge of 100 cm (Figure 5).

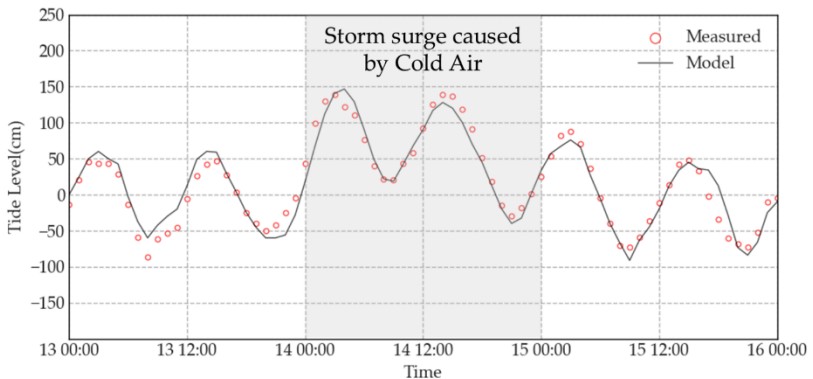

**Figure 5.** Comparison of measured and modeled tide levels during storm surges.

The flow velocity, direction, and total water depth were observed continuously for 27 h from 15 to 16 September 2021, outside of Dongying Port (Point 2) and Gudong Seawall (Point 3). The total water depth variations were measured by a sounding instrument, and the flow velocity and direction were measured with AquaPro, produced by the Norwegian company Nortek, in Vangkroken, with a measurement error of ±0.5 cm. During the observation period, the instrument was fixed to the surface layer and the transducer was kept vertically downward. With a sampling interval of 1 h and a vertical resolution of 1 m, the flow velocity and direction data were averaged vertically for model verification. Figure 6 illustrates the model validation of water depth, velocity and flow direction during normal weather.

### 3.2. Directional Wind Inundation in Different Return Periods

Since the last storm surge in the YRD usually lasts less than two days, only the flooding within 48 h after adding the wind field is considered. The model results are output every 12 h since 0:00 on 1 September 2021. Figure 7 presents the tidal level variation at the YRD at each output time. During the simulation, the YRD is experiencing a shift from neap to spring tides, 12 and 36 h at high tide, and 24 and 48 h at low tide.

The inundation in the YRD is closely related to the astronomical tides, wind field intensity, and topography. As expected, the water depth in the flooded area increases with higher astronomical tides. Therefore, 36 h of coastal flooding was selected to analyze the storm surge in the YRD. A strong link between the intensity and duration of the wind field and the flooding area could be observed; topography determines the location and affected area of the sea overflows. Since the shoreline or the boundary between the delta land and the sea is extracted at low tide level, some bare flats in the south of the Yellow River estuary, which are often inundated at high tide, are included in land areas.

NE winds caused the most extensive flooding. Figure 8 shows the flooding after 36 h of no wind and 50-year N, E, and NE winds. It was found that in N winds, flooding mainly occurs in the coastal concave area of Diaokou Town, which is on the northern side and estuary area of the YRD. The maximum flooding distance of Diaokou Town reaches 28 km, and the maximum inundation depth is 0.9 m. In the estuary, a 270 km$^2$ area was inundated with a maximum inundation depth of 0.6 m. The coastal flooding caused by the E wind

in the 50-year return period mainly occurred in the Yellow River estuary. The maximum coastal flooding distance reaches 30 km, and the maximum flooding depth reaches 0.8 m. The maximum flooding depth of the Gudong Oilfield reaches 0.7 m. Floods caused by NE winds are accompanied by larger inundation areas, coastal inundation distances, and inundation depths than those caused by strong winds in the N and E directions. The coastal flooding caused by the NE strong wind generally occurs on the coast where seawall facilities are absent, especially in the Diaokou town area and estuary. The inundation area of Diaokou town is about 360 km$^2$, and the maximum flooding depth and maximum flooding distance are 2 m and 24.5 km, respectively. The maximum flooding distance of the estuary is 39 km, and the inundation water depth in most areas is greater than 0.5 m, while the inundation depth near the shoreline is generally greater than 1 m.

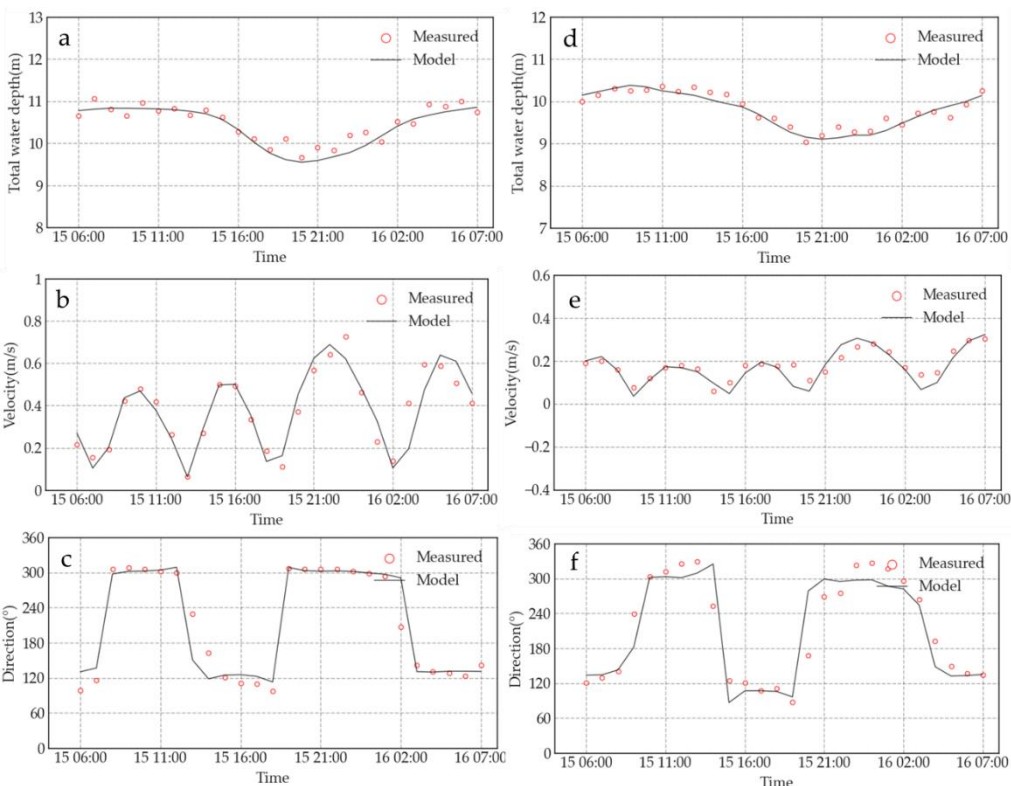

**Figure 6.** Model validation during calm weather period: water depth, wind speed, and direction at Dongying Port station (**a**–**c**) and Gudong Seawall (**e**–**f**).

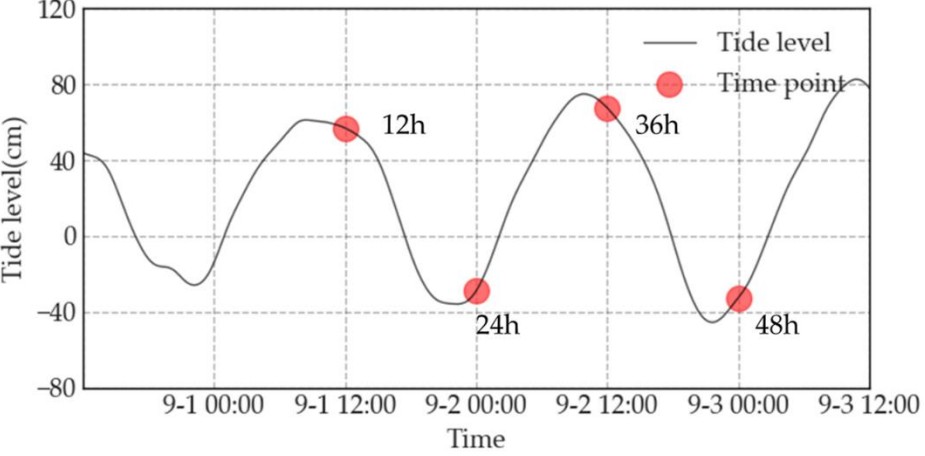

**Figure 7.** Tidal levels in the YRD.

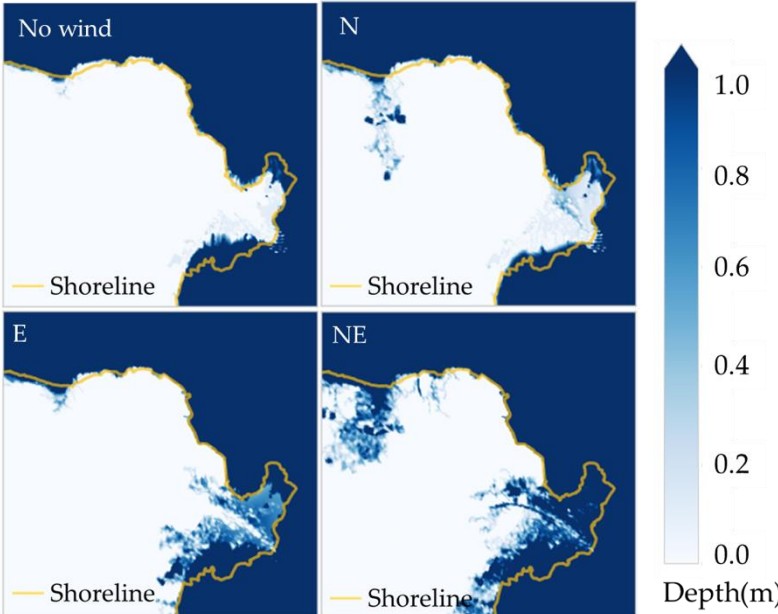

**Figure 8.** Different wind directions in the inundation area for the 36th hour of the 50-year return period, with windless conditions as a control.

The severity of coastal flooding caused by strong wind varies with wind return periods. Figure 9 shows the coastal flooding after 36 h of strong wind in different directions in return periods. According to the above analysis, the wind direction determines the location and area of the flooding.

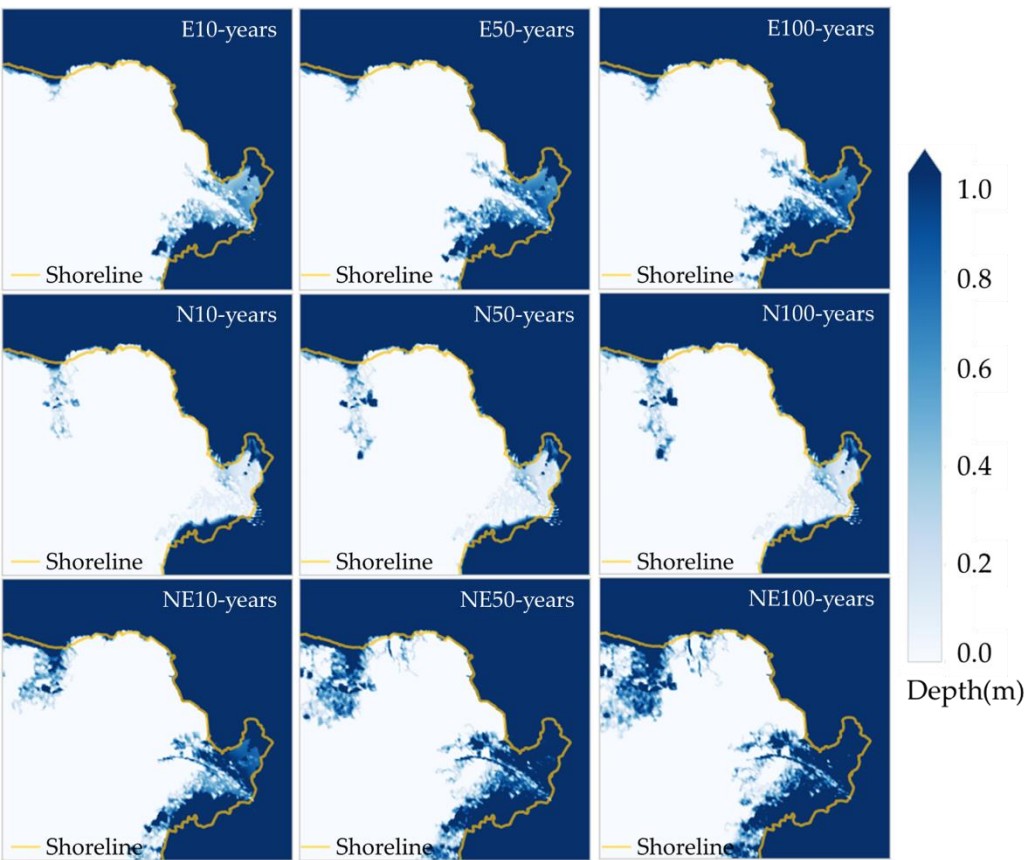

**Figure 9.** Inundation area in the 36th hour of different return periods winds.

In the case of strong E winds, the influence of return periods on the inundation level of the Yellow River Estuary is notable. Under the influence of a 10-year return period E wind, the inundation depth remains below 0.5 m in most areas and reaches 0.5–1 m on the shoreline less than 2 km away. In addition to the area near the Gudong oilfield, the flooding distance in other areas is 6–8 km. The 50-year return period E wind exerts an even greater influence on coastal flooding as the coastal flooding distance generally increases by more than 1 km, especially in some areas of Huanghekou Town and Gudong Oilfield, where the flooding distance increases noticeably and reaches 3–5 km. The rise in inundation depth ranges from 0.1 to 0.3 m in most places to 0.5 to 0.6 m in other regions. As for the 100-year return period conditions, compared with the 50-year return period, there is minimal variation in the distance and area of coastal flooding, and the rise in inundation depth remains under 0.2 m.

Under the different return periods of N wind action, the location of the coastal flooding is relatively fixed, while the coastal flooding distances in Diaokou Town vary significantly in different return periods. The maximum coastal flooding distance during the 10-year return period is 22 km, and the maximum coastal flooding distances during the 50-year return period and the 100-year return period are 30 and 31 km, respectively. In the estuary, the 10-year return period N wind causing the coastal flooding area is about 161 km$^2$, and the same area caused by the 50-year return and the 100-year return N wind, is about 190 km$^2$. Under the influence of a 10-year return period, local low-lying area inundation depth reaches 0.6 m, and other inundation area flooding depth is generally less than 0.2 m. Under the influence of the 100-year return period, the locally maximum inundation depth is 1.4 m. The inundation depth change caused by N wind of different return periods is not significant. When compared to the 10-year return period, the inundation depth increase in the majority of regions throughout the 50-year return period is less than 0.1 m, while in some areas an increase in flooding depth might reach roughly 0.8 m. The difference in inundation depth throughout the 100-year return period is negligible when compared to the 50-year return period.

The YRD flooding is most serious under the control of the NE wind. As for those in the 10-year return period, the flood in Diaokou town reaches a distance of 18 km and inundates an area of nearly 210 km$^2$. In most areas, inundation depth remains below 1 m, while the maximum inundation depth reaches 2.03 m. In the Yellow River Estuary, the maximum flooding distance is 18 km, and the flooding area is 420 km$^2$, with the inundation depth gradually decreasing from the shoreline to inland. In the delta lobe area, the inundation depth is generally 0.5–1 m. Compared with the 10-year return period, the 50-year return period caused a significant increase in distance, flooding area, and inundation depth. Flooding is evident on the north bank of the YRD in the town of Diokou and the west side of Dongying Port. The maximum flooding distance is 27 km in the concave area of Diaokou Town, and the other shorelines' flooding distance is 10–13 km. In the Yellow River estuary, the maximum flooding distance is 22 km, and the inundation depth of nearly half of the inundation area is greater than 1 m. Compared with the 50-year return period, the inundation distance increases slightly for the 100-year return period, and the change range of inundation depth in most areas is less than 0.2 m.

### 3.3. Vulnerability Assessment of Storm Surges

Based on the Sentinel 2 image on 25 July 2021, the land in the YRD was classified according to the land use categories described in Section 3.3, and the classification results are provided in Figure 10. Lands in the YRD are classified into four main types, namely industrial and residential lands, bare tidal flats, water, and vegetation. Dongying Port and Gudong Oilfield are mainly used for industrial purposes, and residential land in the delta is mostly aggregated as clumps. The bare tidal flat is mainly distributed in coastal areas, and water surface is concentrated mainly in the east of Diaokou Town and Xianhe Town. In the YRD, cultivated land vegetation is widely distributed.

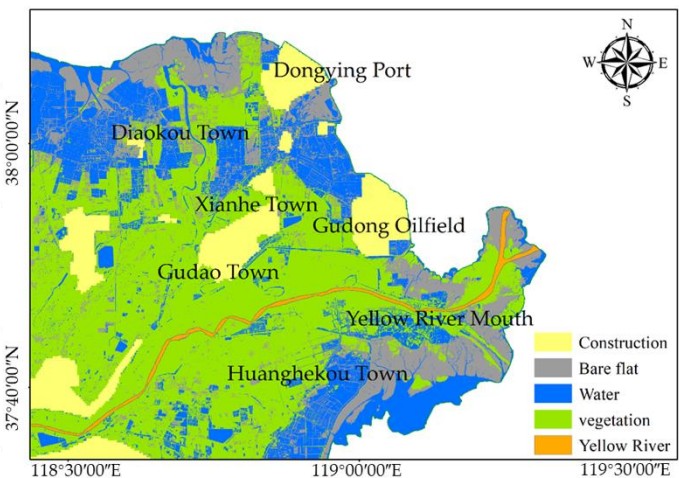

**Figure 10.** Land use in the YRD (July 2021).

Major industries in the YRD include petrochemicals, salt production, and aquaculture. Dongying Port and Gudong Oilfield, the hubs of the petrochemical industry, where the relevant storage and transportation equipment are located, face a higher ecological risk of flooding damage and the highest vulnerability level (level I). The salt industry and aquaculture are widely distributed on the nearshore, which is part of the agricultural land of the facility, and the vulnerability level is set at a lower level (level III). By mapping the assessment results of vulnerability against storm surges across different regions in the YRD, as shown in Figure 11, it was found that urban residential areas, Dongying Port Petrochemical Industry Zone, and Gudong Oilfield Production Zone are at level I vulnerability; aquaculture and salt production pond water area are at level III vulnerability; and the bare flats and cultivated land vegetation areas are at level IV.

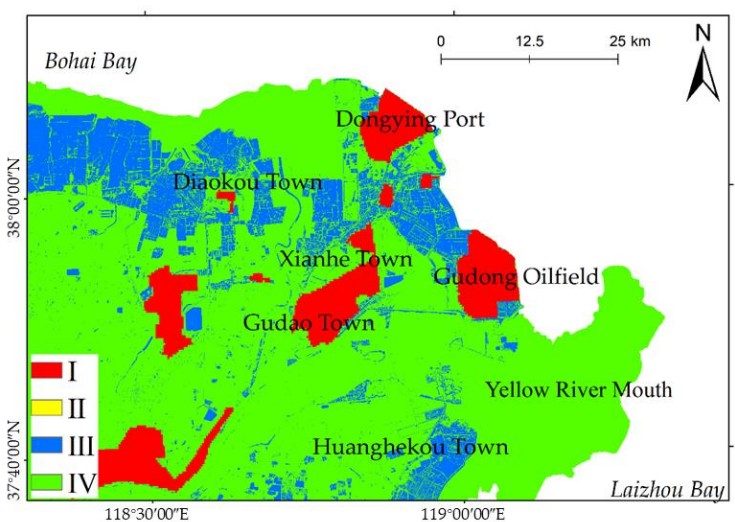

**Figure 11.** Vulnerability levels of storm surge in the YRD.

*3.4. Storm Surge Risk Levels*

As described in Section 3.2, floods caused by NE winds have a wider impact and greater depth than those caused by N and E winds. Hence, floods caused by NE wind are selected as the rating of storm surge risk level in the YRD.

The overall risk level of storm surge caused by NE strong winds in the YRD is low, with the majority of areas at lower risk (level III) and low-risk areas (level IV). Even with strong winds in the 100-year return period, the risk area of level II is only 12 km$^2$. The estuary and the town of Diaokou are prone to risk. Diaokou Town appears to be greatly

affected by the strong wind in different return periods, as the 10-year and 100-year return period of winds, respectively, result in 261 and 640 km² of risk areas. It is worth noting that in a 100-year return period of strong wind, the south of storm surge risk area is very close to the urban area of Hekou town, even imposing a lower-risk area of storm surge (level III) in the northern part of the urban area (Figure 12).

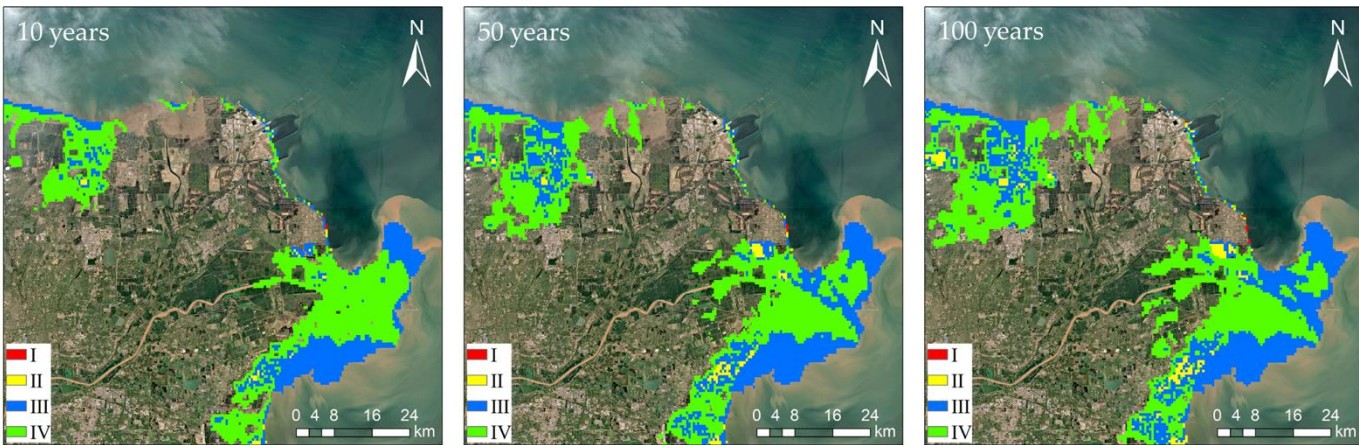

**Figure 12.** Risk map showing risk level of storm surge across the YRD in different return period winds.

## 4. Discussion

### 4.1. Risk of Storm Surge to Oilfields

The coastal oil fields are located in intertidal and shallow water areas for oil and gas extraction. At present, more overflow roads and artificial islands are being constructed to realize offshore oil mining on land. Since the construction of the Gudong Oilfield in 1985, 10 coastal oilfields have completed construction in the YRD, such as WHZ, XT, and ZX (Figure 13). Due to frequent storm surge disasters [6] and shoreline erosion [23], coastal oilfields are exposed to greater risk of storm surge, while other facilities such as overflow roads, oil production platforms, seawalls, and revetments are seriously damaged.

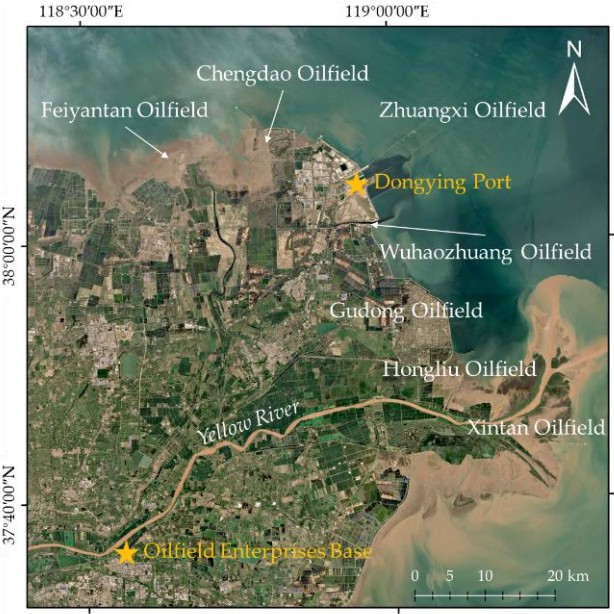

**Figure 13.** Coastal oilfields in the YRD, including Feiyantan (FYT), Chengdao (CD), Zhuangxi (ZX), Wuhaozhuang (WHZ), Gudong (GD), Hongliu (HL), and Xintan (XT) oilfields.

Figure 14 shows risk levels of storm surges caused by 100-year return period winds in different directions. The risk level of storm surge in coastal areas remains high in all scenarios (level I), especially in NE and E winds. However, for windless and N-directional winds, the storm surge risk level is relatively low (level II) at 1.5–2 km from the shoreline. The risk level is even lower near HL Oilfield, and the area within 3 km from the coastline is determined to be below level II.

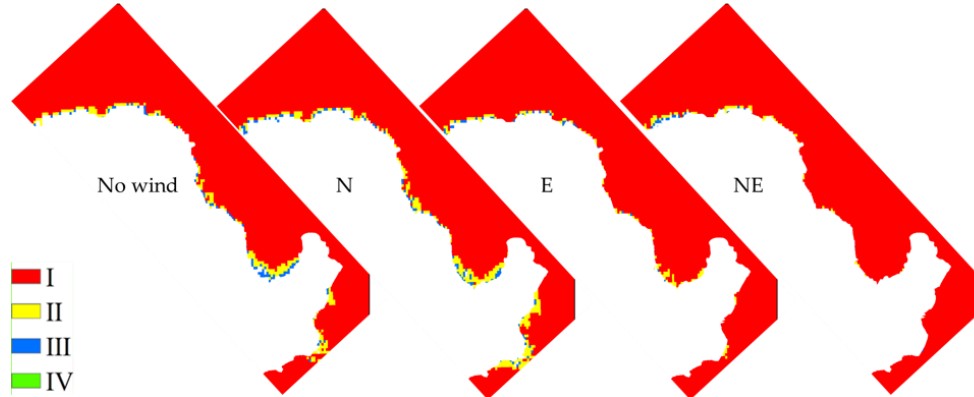

**Figure 14.** Storm surge risk levels in the YRD with 100-year return period winds in different directions.

The storm surge risk level along the YRD coast varies under different conditions, which is mainly related to the rise and fall of the sea surface during the storm surges. Wang et al. (2017) simulated the storm surge for different wind directions in the YRD and found that strong wind can lead to large changes in water levels [7]. For coastal oilfields, water level changes are strongly associated with storm surge risk.

Figure 15 shows the YRD storm surge for 36 h of 100-year return period wind. Under the control of N winds, the water level in the YRD increases from north to south, except for certain sites on the north estuary and the incurvate shoreline in Diaokou Town, where the storm surge reached roughly 0.5 m. Water levels did not alter substantially in other places. On the south side of the estuary, the water level decreased by about 0.6 m. Under the control of E winds, the water level consistently increased by about 0.9 m. Under the control of NE wind, the storm surge generally reached 1.25–1.5 m. HL Oilfield and WHZ Oilfield had storm surges of over 1.5 m. Compared with the N and E directions, the NE wind causes storm surges, which is more threatening to the safety production of coastal oilfields.

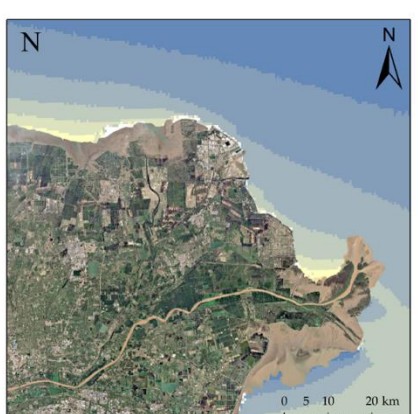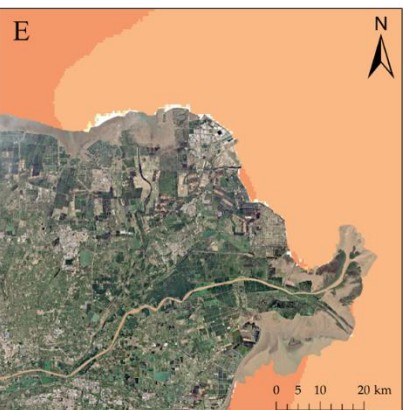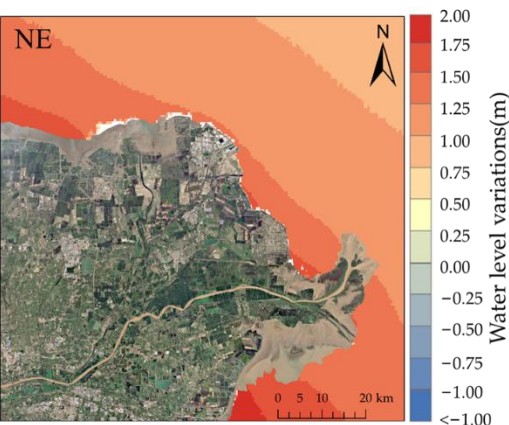

**Figure 15.** The YRD water level rises to control the 100-year winds in different directions.

### 4.2. Morphological Evolution and Storm Surge Process

River flow, wave energy, and tidal range are key factors in the morphological evolution of the delta [9]. However, in recent years, sea level rise, insufficient sediment supply, human intervention, and climate change have continued to lead to frequent floods and storm surges,

which have now emerged as new factors in remodeling large deltas [24,25]. In the YRD, the impact of storm surge on topography is mainly manifested in the erosion of the Gudong nearshore and its northern region [9]. However, for tidal flats with high critical bed shear stress, erosion of storm surges is not obvious, and even siltation occurs in some regions [8]. Geomorphic changes will cause local hydrodynamic changes, which in turn will affect the storm surge.

Based on the YRD underwater cross-section measurements collected in 1985, a simulation of a 100-year storm surge under the influence of different wind directions was performed to compare with the simulation results of topographic conditions in 2015 (Figure 16). Compared to 1985, erosion in the YRD mainly occurred outside of Gudong oilfield and its northern area, while accretion mainly occurred in the active estuary. Therefore, the increased water depth in the northern part of the Gudong Seawall and the YRD confirms the increased risk of storm surge in the coastal oilfields.

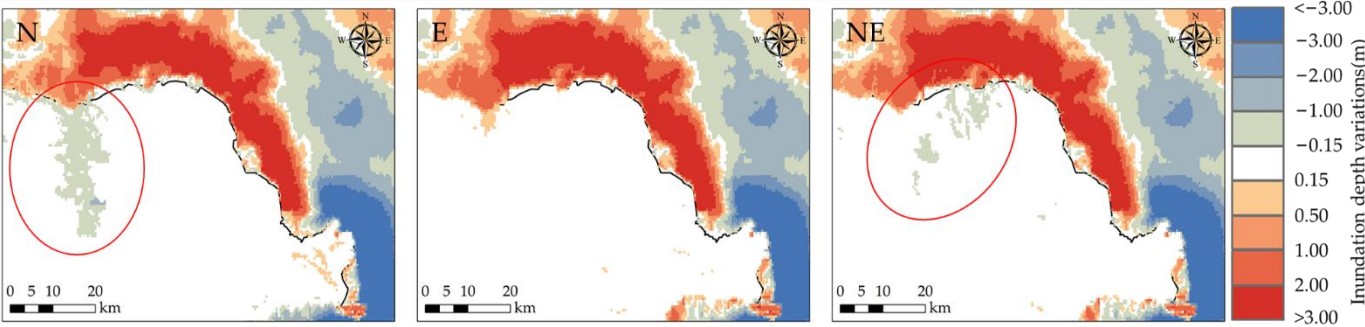

**Figure 16.** Inundation depth variations in 100-year return period wind in different directions.

The morphological evolution also changed the storm surge pattern of the YRD. In general, Diaokou Town in the northern YRD saw the greatest shift in the storm surge flooding pattern in 2015 when compared to the topography in 1985. In addition, the flooding distance and inundation depth for the 2015 topography were weaker than those for the topography in 1985. When under the control of N wind, the inundation depth in the concave area of the Diaokou Town shoreline is reduced by 0.15–1 m, and the coastal flooding distance is reduced by 9 km. When under the control of NE wind, a 0.15–1 m decrease in the inundation depth on the west side of Dongying Port could be observed with a fixed area coverage. However, the inundation depth at Diaokou Town's shoreline increased by 0.2 m while the E wind was in control, which may be due to the erosion of abandoned deltas in the north.

The bathymetric data of three distinctive cross-sections outside of Diaokou Town in the northern YRD were investigated to determine the reasons for a decline in coastal flooding (Figure 17). The selected locations of the characteristic sections are on the incurvate shoreline with a gap of 15 km. In contrast, Line 1's topography has not changed significantly, while Line 3's erosion was more severe between 5 and 14 km from the coast. The depth of erosion at 6 km is 3.8 m, and the offshore underwater slope is significantly steep. The section erosion of Line 2 is the most serious, with an average of 1.7 m. It can be seen that the flooding depth of the land area at the position of Lines 2 and 3 is less than the change in the inundation depth of the storm surge depicted in Figure 16. The reason may be coastal erosion reshaping the bathymetric profile, making it more difficult to climb to the land due to the increase in slope.

### 4.3. Storm Surge Disaster Defense

Storm surge is one of the most serious natural disasters in coastal areas, and how to control storm surge damage has always been a public concern. Over past decades, 'gray sea dykes', hard-structure buildings of steel, concrete, dolosse, and stone, have remained the major form of coastal defense. Due to high construction costs, delta land subsidence [26,27], sea level rise [28], and human activities, 'green sea dykes', which

combine traditional engineering structures with eco-systems to make full use of wetlands, reefs, and other providers of ecosystem services, have gained more popularity among researchers in recent years [29]. Compared with the 'gray seawall', coastal ecosystems such as beaches, mangroves, and swamp wetlands are more effective in wave dissipation and sediment storage.

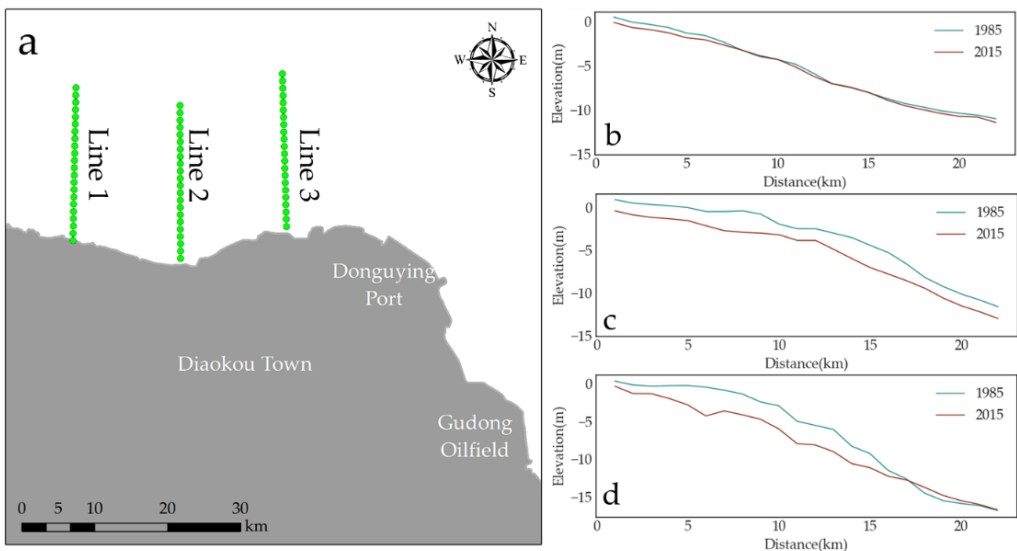

**Figure 17.** Bathymetric profiles of three cross-sections at Diaokou. ((**a**): Locations of cross-sections; (**b**): Line 1; (**c**): Line 2; (**d**): Line 3).

At present, the 'gray sea dyke' is the basis of coastal protection projects in the YRD, lining the coast from Gudong Seawall to Dongying Port. Experience has shown that these seawalls could effectively block the movement of seawater toward land, yet the storm surge often led to the collapse of the seawall slope protection and retaining wall, the loss of rubble, the rolling of dolosse, and the collapse of overflow roads. Seawalls are carved by the strong seabed erosion, giving them an inverse profile shape of 'nearshore deep, offshore shallow'. It is reported that more than 84% of the seawalls are in a seriously unstable state [30]. In addition, due to the tectonic subsidence of loose sedimentary strata and the natural consolidation of sediments in the modern YRD, as well as human activities such as groundwater withdrawal, engineering construction, and oil and gas exploitation [31], the whole YRD region is experiencing land subsidence, with a maximum vertical subsidence rate of 432 mm/y [17]. Consequently, the seawall will deform, and its protective capacity will decrease year by year. It is expected that by 2050, overflows will occur in the Gudong Seawall [15].

The main wetland types in the YRD are water, tidal flat, herbaceous swamp, and shrub swamp, and the dominant plant species are *reed*, *Suaeda salsa*, and *Spartina alterniflora*. *Spartina alterniflora* is an important land-silting and wave-sweeping revetment plant that mainly grows in the middle and lower intertidal zone. In the Yellow River Estuary, its growth area increased from 221.85 hm$^2$ in 2002 to 5267.79 hm$^2$ in 2020 [32]. With strong survival and growing abilities, it has great potential to prevent storm surge disasters. In addition, large tidal flats in the YRD, where storm surges frequently occur, are well suited for the growth of salt marsh vegetation such as *Spartina alterniflora*. However, the risk of species invasion should be considered when growing alien species such as *Spartina alterniflora*.

In the future, the construction of seawalls and other protective facilities would remain the major task in storm surge prevention, with an upgraded prediction system of storm surge disasters in place. Along the coastline from Gudong Oilfield to Dongying Port where the erosion of the seabed in front of the seawall steals space for the growth of salt marsh vegetation, it is necessary to enhance the routine inspection and maintenance of the seawall.

When seabed erosion is spotted near seawalls, timely repair and consolidation should be ensured. For other areas, 'gray seawalls' and 'green sea dykes' should be reasonably adopted in flooding sites, and salt marsh vegetation should be used to dissipate wave energy to ensure the safety of industrial and agricultural production.

## 5. Conclusions

In this study, a numerical model, aided by satellite and measured data, was used to simulate the storm surge in the YRD under different conditions. Based on the coastal floods caused by the NE wind in the 100-year return period, the risk level of storm surge in the YRD is specified. The following conclusions are drawn.

(1) The YRD is most severely affected by the NE wind. The maximum flooding distance of NE wind in the 10-year return period is 18 km, and the inundation areas in Diaokou Town and Yellow River Estuary are 210 and 420 $km^2$, respectively. However, the overall storm surge risk is at a low level in different return periods of strong winds. In the 100-year return period of strong winds, the higher risk (level II) area is only 12 $km^2$. The storm surge risk areas were mainly distributed in the Yellow River Estuary and Diaokou Town.

(2) The coastal oilfields along the YRD are principally at a high storm surge risk (level I). Strong winds can cause changes in sea level. Under the control of E and NE winds, the water levels along the delta coast increase by 0.9 and 1.4 m, respectively, thereby increasing the storm surge risk in coastal oilfields. However, under the control of N wind, the change of water level along the coast is negligible, with a maximal decrease of 0.6m on the southern side of the Yellow River estuary.

(3) Topographic changes affect the pattern of storm surge flooding over the YRD. When seabed erosion significantly deepens the water in the north and northeast of the YRD, the land flooding distance is reduced by 9 km, and the inundation depth is also reduced by 0.15–1 m, under the control of N wind and NE wind.

(4) Seawalls and other protective projects are crucial for storm surge defense in the YRD. On the basis of an upgraded prediction system of storm surge disasters, it is necessary to better maintain seawalls and reduce seabed erosion along the coastline from Gudong Seawall to Dongying Port. For other coastal areas, a combination of the salt marsh vegetation such as *Spartina alterniflora* facilities would yield better resistance against storm surge disasters when combined with traditional defenses.

**Author Contributions:** Conceptualization, L.H. and S.C.; methodology, L.H. and S.P.; software, L.H. and H.J.; validation, L.H. and P.L.; formal analysis, L.H. and S.C.; investigation, L.H.; resources, S.C.; data curation, L.H.; writing—original draft preparation, L.H.; writing—review and editing, L.H. and P.L.; visualization, L.H.; supervision, S.C.; project administration, S.C.; funding acquisition, S.C. All authors have read and agreed to the published version of the manuscript.

**Funding:** This study was partly supported by the National Natural Science Foundation of China (NSFC, No. U1706214) and the Open Research Fund of SKLEC (SKLEC-KF202001).

**Data Availability Statement:** The data that support the findings of this study are available from the corresponding author upon reasonable request. The following supporting information can be downloaded at: Figure 1b: Delta land elevation, https://www.gscloud.cn/ (accessed on 20 April 2022); Figure 2: Annual distribution and cause of storm surge in the YRD from 2014 to 2021, http://hsdy.dongying.gov.cn/col/col36603/index.html (accessed on 20 April 2022); Table 1: Topo-graphic out-side of Bohai Sea, http://mds.nmdis.org.cn/pages/dataView.html?type=1&id=4cbe083006234f51a4b2782368b3f38f (accessed on 20 April 2022).

**Acknowledgments:** We would like to acknowledge the hydrological dataset provided by the Yellow River Conservation Commission, the Ministry of Water the Resources of China and all open-source data providers (the providers mentioned appear in the article). We also thank anonymous reviewers for their constructive comments.

**Conflicts of Interest:** The authors declare no conflict of interest.

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
