# Peer review of "Impact of Storm Surge on the Yellow River Delta: Simulation and Analysis"

_water, doi:10.3390/w14213439_

Round 1

Reviewer 1 Report

This paper modeled the impact of storm surge on the YRD and assess the risk. Possibly it has the potential to be published, but not the present form.

Major comments:

1. The writting is a little difficult to understand for an international reader. At some places the expression is unclear, or illogicalr.  It's better to ask for help from an international expert or a native english speaker. 

2. The bathymetries in 1985 and 2015 have been used to test the influence of topography. However, only three sections were illustrated. The other parts of the bathymetries should be added, because in the paper bathymetric changes in other parts of the delta have been mentioned. Moreover,  it is also necessary to discuss the impacts of the future topographic evolution of the delta on the storm surge.

3. The present study considered the impacts of the directions and return periods of strong winds on the storm surge. In my opinion, the astronomical tides can influence the storm surge, too. For example, the storm surge during a spring tide would be stronger than during a neap tide. This must be treated reasonably. 

4. The methods for risk and vulnerability assessment were from the guideline published by Ministry of Natural Resources of China in 2015.  Tables 3, 4, 5 also from this guideline. The guideline method can be described more simply while the other updated methods can be reinforced.

Minor comments (not limited to these examples, the authors should improve the paper thoroughly):

1. How did you extracted the tidal boundary conditions from the software?

2. Place names should be added in  figures7, 8,11, 13, 14, 15.

3. Sentinel image should be added to table 1.

4. The cell size of the model should be described, including the minimum and maximum size.

5. References 5 and 18, pls add "in chinese with english abstract".

6.  Which station is the tidal level in Figure 6 from?

7. Give a value of land subsidence in line 431.

8. Line 125, which version of Mike 21 were used? Dhi in line 539 should be DHI.

9. line 474, "," -->"."

10. Line 365, Landform-->morphological.

Author Response

Dear reviewer:

Thank you very much for your comments. In the attachment, some of our thoughts and responses to the comments in it. 

We are looking forward to your comments.

Reviewer 2 Report

Please revise the manuscript according to the attached file.

Author Response

(The authors gave the same response as above.)

Reviewer 3 Report

This paper is written clearly, and it may be interesting for publication in Water.  The main suggestion is connected with the description of the study area. This description is usually description in hydrological and hydrogeological sense (without wikipedia information). Please, include the topgraphic and hydrogeologic maps with the basic data and also the basic statistics of meteorological and hydrological data.  Also in 2.2 Data collection and pre-processing include the graphical presentation of the rainfall through the analyzed period (also average, maximum and minimum values). The description of the storm is also missing. I suggest moderate revision. 

Author Response

(The authors gave the same response as above.)

Round 2

Reviewer 1 Report

The ms has been largely reinforced according to the reviewers' comments. The main problem is still the writing. Here I give some examples, (not limited to them): 

1.line 32, hit--hits

2. Table 2, also add the reference [22].

3. line 191, give the full name of RBR

4. line 376, morphological--Morphological

5. line 65, 'give' shouldn't be 'given' here.

6. line 88, South--south.

7. line 86, with -- being?

8. line 92, images--image, because you show one satellite image here. Also  provide the source of the image. 

9. In the first paragraph of the Conclusions, the author mentioned "based on the coast floods ...by the NE wind in 100-year return,...", but in the second paragraph,  mentioned the "10-year returen period", also results of E and NE winds...

10. line 406-416 should be appear earlier, before the model results of storm surge pattern. why not merge it into line 386-392? There are many places with similar logical confusion.

Author Response

Dear reviewer, thank you very much for reviewing my paper again and giving us lots of valuable advice. Follow your opinion, we checked the article again, and revised many mistakes. As for the following two questions, our opinion is that.

9. In the first paragraph of the Conclusions, the author mentioned "based on the coast floods ...by the NE wind in 100-year return,...", but in the second paragraph,  mentioned the "10-year returen period", also results of E and NE winds...
Re: based on the coast floods under 100-year return wind, we draw the risk map of storm surge. While mentioned the "10-year return period" wind, it is to say flooding inundation situation. These are two problems.

10. line 406-416 should be appear earlier, before the model results of storm surge pattern. why not merge it into line 386-392? There are many places with similar logical confusion.
Re: It must be pointed out that we discuss the morphological evolution based on the discovery of flood response to morphological; therefore, the location of line 406-416 is suitable.

Reviewer 2 Report

The manuscript is acceptable in present form.

Author Response

Dear reviewer, thank you very much for reviewing my paper again and giving us some valuable advice. 

Reviewer 3 Report

The paper may be published in present form.

Author Response

(The authors gave the same response as above.)
